# Influence of Fabrication Method and Surface Modification of Alumina Ceramic on the Microstructure and Mechanical Properties of Ceramic–Elastomer Interpenetrating Phase Composites (IPCs)

**DOI:** 10.3390/ma15217824

**Published:** 2022-11-06

**Authors:** Paulina Kozera, Anna Boczkowska, Krzysztof Perkowski, Marcin Małek, Janusz Kluczyński

**Affiliations:** 1Faculty of Materials Science and Engineering, Warsaw University of Technology, Woloska 141, 02-507 Warszawa, Poland; 2Łukasiewicz Research Network, Institute of Ceramics and Building Materials, Ceramics and Concrete Center, Postępu 9, 02-676 Warsaw, Poland; 3Faculty of Civil Engineering and Geodesy, University of Technology, Gen. Sylwestra Kaliskiego 2, 00-908 Warsaw, Poland; 4Faculty of Mechanical Engineering, Military University of Technology, ul. Gen. Sylwestra Kaliskiego 2, 00-908 Warsaw, Poland

**Keywords:** IPCs, ceramic preform, ceramic–elastomer composite, silane coupling agent, hot isostatic pressing (HIP), wettability, mechanical properties

## Abstract

The paper presents experimental results of the work conducted to improve the adhesion between alumina ceramics and urea-urethane elastomer in the interpenetrating phase composites (IPCs), in which these two phases are interpenetrating three-dimensionally and topologically throughout the microstructure. Measurements of the contact angle, surface roughness, and shear tests were used to evaluate the effectivity and select the quantity of a silane coupling agent and the ceramic fabrication method. The tests were conducted using samples of dense alumina ceramic obtained by three- or four-step methods. In the four-step process, hot isostatic pressing (HIP) was applied additionally. As a result of the coupling agent coat and HIP application, the ceramic substrate wettability by the elastomer was improved. The water contact angle was reduced from 80 to 60%. In the next step, porous ceramic preforms were fabricated using HIP sintering and a solution of silane coupling agent treated their surface. The composites were produced using vacuum-pressure infiltration of porous alumina ceramics by urea-urethane elastomer in liquid form. The influence of the coupling agent application on the microstructure and mechanical properties of the composites was estimated. The microstructure of the composites was identified using SEM microscopy and X-ray tomography. As a result of using the coupling agent, residual porosity decreased from 7 to 2%, and compressive strength, as well as stress at a plateau, increased by more than 20%, from 25 to 33 MPa and from 15 to 24 MPa, respectively, for the composites fabricated by infiltration ceramic preforms with 40% of porosity.

## 1. Introduction

Composite materials are a widely used group of materials in various industries. Traditional composites can be classified according to the shape and geometry of the reinforcement (such as particles, long fibers, short fibers, etc.). Interpenetrating phase composites (IPCs) have become a new group of materials that can be used as light materials with improved physicochemical, mechanical, and thermal properties. The IPCs are characterized by the microstructure with two three-dimensional and topologically interpenetrating phases. Furthermore, the absence of the preferred orientation of matrix and reinforcement in IPCs makes them predominantly isotropic and with consequent properties that are unattainable by other materials [1,2].

The strength of the phase separation surface, called the interphase boundary, depends on the type of phase joint, the appropriate wettability of the solid material by the liquid phase, and the surface topography of the solid component. Regardless of the type of reinforcement, the adhesion between inorganic and organic components in composites is usually weak due to the poor compatibility of the polymer or metal with the mineral surface. The problem of the insufficient wetting of the ceramics by molten metal or polymer is one of surface tension and surface quality, including any contamination, or oxidation. Some basic ways can be used to improve wetting. Generally, increasing the surface energies of the solid, decreasing the surface tension of the liquid matrix, as well as decreasing the solid–liquid interfacial energy at the interface can be essential [3]. Several treatments could be performed to increase the wettability of IPCs. In the work [4,5] it was found that the pressure-assisted infiltration, as well as process temperature and gas atmosphere, overcomes insufficient wettability between liquid metal and ceramic preform compared to pressureless fabrication methods. Moreover, in the case of pressureless techniques, insufficient wettability of the ceramic preform by liquid metal causes the formation of intermetallics as a result of interfacial reactions. According to [6,7], the application of some additives to aluminum alloys used in pressureless infiltration to the fabrication of ceramic–metal interpenetrating phase composites improved the wettability between the metallic and ceramic phases. Wang et al. [6] studied the microstructure of interface in 3D-SiC/Al-Si-Mg interpenetrating composites. It was shown that to improve wettability; the optimum content of Mg addition to the Al alloy was 4–8 wt%. A similar effect was obtained for SiC/aluminum alloy composites fabricated by pressureless infiltration as a result of the addition of 6–12 wt% Si [7]. A new method of producing ceramic-metal IPCs is the focus of the work reported by Qi et al. [8]. In the ultrasonic infiltration, the generated ultrasonic waves cause a collapse of gases dissolved in the melt and those entrapped in the ceramic foam. As a result, a pressure wave provides filling of the porous ceramic preform by the molten alloy. Furthermore, it was observed that the contact angle ZrB_2_–SiC porous ceramic by aluminum alloy decreased, improving wettability.

An alternative possible technique to improve adhesion between composite phases is to modify the filler surface with a surfactant, a coupling agent, or other surface-active agents. They reduce surface tension at the interface, which has a positive effect on the adhesion between composite components [9,10]. In the literature, the application of an organosilicon coupling agent, especially for composite materials based on ceramics and polymers or metal and polymer, is widely described. When an organofunctional coupling agent is added to ceramic–polymer composites during the polymerization or crosslinking process, it forms a phase at the boundary by reacting with the functional groups of the polymer [10,11,12]. The use of coupling agents can increase the degree of ceramic pore filling by the polymer in a liquid form up to 80 vol.%. In addition, improving the adhesion between the components of the composite increases its compressive and tensile strength. Moreover, it can also lead to increased abrasion and corrosive resistance of the composite [11,13,14].

In polymer ceramics composites, microcracks may induce fracture to occur during stressing at or near the components’ boundary. Therefore, the endurance of the connection between two phases in a composite is one of the main factors determining the IPCs’ properties. To improve the mechanical properties of inorganic–organic phases, coupling agents can be applied. In the work [15], it was found that using silane coupling agents may cause the formation of siloxane bridges at the interphase, and as a consequence, result in the improvement of adhesive strength of polymer to ceramics.

In the work of [16], the effect on the dielectric and mechanical properties of co-continuous barium strontium titanate (BST)–polymer composites of surfactant introduction onto the ceramic filler surface was investigated. The result means that carboxylic acid with longer carbon chains enhances the relative permittivity of the composite more effectively. The feasibility of processing polymer syntactic–aluminum foam IPC and its compression response was studied by Jhaver and Tippur [17,18]. They applied silane to increase adhesion between the metal scaffold and polymer foam. The IPC foam coated by silane showed improvement in elastic modulus, compression strength, and plateau stress values by 28–35%, 20–25%, and 37–42% respectively.

The adhesion of two phases in IPC composites depends on the development of the ceramic surface. The topography of the ceramics**’** surface is changed according to the method and parameters of fabrication. Generally, during the sintering of a powder compact, both densifications occur simultaneously. The most effective way to stop grain growth and, as a result, obtain a ceramic fine-grain structure is to support the sintering processes with high pressure. Despite the high cost of the Hot Isostatic Pressing (HIP) technique, most ceramics, sintered under isostatic pressure gas, compared to freely sintered materials, are characterized by better fracture toughness [19], increased compressive strength, and tearing [20,21], as well as greater Vickers hardness [22]. Taking into consideration the porous ceramic fabrication, the HIP technique is a relatively rarely used method. Polymeric sponge method or polymer replica technique, gel casting of foams of ceramic foams all well as porous structure printing are more commonly used methods. Regardless of the method used and the pore size achieved, fabrication of a preforms structure of open pores and joined canals is required in the case of IPC materials. This structure allows for the easy flow of the liquid material.

In work [23], the authors present a comprehensive description of the freeze casting method, also known as ice templating, which has been applied extensively for the fabrication of well-controlled biomimetic porous materials based on ceramics, as well as metals. Infiltration of freeze-cast skeletons with a secondary phase allows for fabrication of aligned composites, hydrogels, and nacre-mimetic hybrids, as well as materials resembling interpenetrating phase composites.

This paper presents the influence of the fabrication methods of ceramic and its surface modification by the coupling agent application on the microstructure and mechanical properties of the ceramic–elastomer composite. Tests were carried out to obtain an accurate ceramic–elastomer joint characterization, measurements of a contact angle, surface roughness, and shear strength. The appropriate amount of coupling agent was determined for ceramic–elastomer composite fabrication. The application of a silane coupling agent improved the properties of the composites, as well as a ceramic–elastomer joint.

## 2. Materials and Methods

### 2.1. Materials

Two types of alumina ceramic samples were fabricated. The first type, dense ceramic specimens, investigated how the hot isostatic pressure sintering application affects ceramic surface roughness. Then, the dense ceramic specimens were treated with the silane coupling agent and measurements of the contact angle, and shear tests of adhesive joints were carried out. In the second type of sample, porous ceramic preforms were applied to ceramic–elastomer composites fabrication and then was analyzed how the coupling agent affected their microstructure, residual porosity, and compressive strength as well as stress at the plateau. Moreover, porous ceramic preforms were tested using X-ray tomography and a strength testing machine. All the material fabrication processes carried out are shown schematically in Figure 1.

The dense alumina ceramic samples were prepared by the three-step or four-step methods. First, the pre-compaction process of Granulox NM9922 (Nabaltec AG, Schwandorf, Germany) alumina granulate in steel molds by uniaxial pressing at a working pressure of 200 MPa was conducted. Secondly, the samples were densified into green bodies by cold isostatic pressing (CIP) under a pressure of 200 MPa for 1 min. Next, the free sintering process (pressureless sintering) was carried out in a Nabertherm HT 08/18 electric chamber furnace (Nabertherm, Lilienthal, Germany). An air inert atmosphere was used. After reaching 1600 °C, the samples were heated for 2 h to get rid of organic binders contained in the pressed shape. The samples produced by the three-step method were designated Al_2_O_3_-SS. A part of the samples marked Al_2_O_3_-HIP200 was additionally treated by HIP process under a pressure of 200 MPa in an argon atmosphere at 1600 °C for 1 h.

The urea–urethane elastomers (PU2.5) were synthesized by a one-shot method from 4,4′-methylenebis(phenylisocyanate) (MDI), poly(ethylene adipate) PEA, and dicyandiamide (DCDA). The molar ratio of MDI to (PEA + DCDA) substrates was equal to 2.5 (which means a hard to soft segments ratio equal to 1.50).

The porous ceramic preforms applied for the infiltration process were fabricated using the same methods as dense ceramic samples. However, to obtain open porous preforms, the alumina powders supplied by the P.P.U.H.KOS company were mixed with 10–15 wt% of Granulox NM9922 (Nabaltec AG, Schwandorf, Germany) high-temperature ceramic binder and 7 wt% of dextrin solution as pore structure forming agent. Two different sizes of alumina powders were used to fabricate ceramic preforms with different values of porosity. As a result of an application of alumina powder with 300–100 µm and 1200–1000 µm size, preforms with 20% and 40% porosities were produced, respectively. After the preparation of two types of ceramic mixtures, they were inserted into steel forms and molded by uniaxial pressing at a working pressure of 100 MPa. In the next step, the pre-sintering of semi-finished preforms was performed using an electric chamber furnace. Finally, to enhance mechanical properties and increase the density of ceramic preforms, hot isostatic pressing was conducted under 200 MPa pressure in argon at 1600 °C temperature for 1 h.

The ceramic–elastomer composites were made by the infiltration of the ceramic preforms with 20 vol% and 40 vol% porosity by a reactive mixture of urea–urethane elastomers in the liquid form. The infiltration was carried out under subatmospheric pressure.

The coupling agents can be used in two ways. In general, the surface of ceramics before infiltration by a polymer is covered by the solution of the coupling agents [11,14]. It was claimed that the most satisfying results were achieved when the agents were dissolved in benzene, toluene, and alcohol in an amount of not more than 10% by weight. In the case of ceramic–polymer composites, to cover the pore surfaces thoroughly, it is recommended to apply pressure pre-infiltration of the ceramic preforms by a coupling agent. Then, the process of drying the preform is used. This type of sample preparation is characterized by high durability and resistance during storage. The disadvantage of this method is the need to introduce into the fabrication process an additional step consisting of coating the ceramic preform with a coupling agent. Another way of applying the coupling agent is by adding it to the reactive mixture of polymer before filling the ceramic pores. During infiltration, the coupling agent migrates to the phase boundaries, acting as an adhesion promoter. The advantage of this method is the ease of implementation. However, there are several problems regarding the stability of the agent and the durability of its conjunction with the polymer, due to the presence of moisture in the system. Moreover, it was observed that a higher amount of coupling agent should be used, because a significant part of it does not reach the interface and remains in the “mass” of the polymer [10].

In this work, a Unisilan U-15 silane coupling agent (UNISIL Company, Tarnów, Poland) was applied. The U-15 Unisilan with the chemical name N-2-aminoethyl-3aminopropyl-trimethoxysilane is mainly used as a coupling agent to inorganic fillers in composites based on acrylic polymer, epoxy resin, vinyl polymer, and polyamide, polyether, polyurethane, silicone. A U-15 silane coupling agent was applied directly to the ceramic surface. Dense ceramic samples, as well as porous ceramic preforms, were immersed in the solution of 1 wt% and 5 wt% U-15 agent in toluene for 5 s. Next, the samples were placed in an oven for 2 h at 120 ± 5 °C under a vacuum to evaporate the toluene. Next, the temperature was reduced to about 80 °C and the samples were heated for another 18 h.

### 2.2. Methods

The roughness tests were performed using a non-contact 3D surface profiler Slynx Sensofar (Sensofar Metrology, Terrassa, Spain). Two parameters were determined: the R_a_ parameter, i.e., the arithmetic mean height of the surface, and the R_z_ parameter, i.e., the height of the highest point on the surface. The final values are the average of three different measuring points on the surfaces.

The wettability of the ceramic surfaces was tested by measuring the water contact angles (WCA) and the contact angles hysteresis (CAH) using an OCA15 (DataPhysics Instruments, Germany) goniometer equipped with OCA software. The contact angle hysteresis was determined by calculating the difference between the advancing and receding contact angles. The test was performed on rectangular tiles with dimensions of 50 *×* 40 *×* 10 mm using a 5 μL droplet. The final WCA values are the average of three different measuring points on the surfaces.

To examine the strength of the adhesive ceramic–elastomer joints, a shear test was conducted. The Lloyd LR 10K (AMETEK, Berwyn, IL, USA) strength testing machine connected with the Nexygen 3.0 computer program (AMETEK, Berwyn, IL, USA) was used for the tests. Shear strength was calculated from the force registered by the computer divided by the joint cross-section according to the formula (1). The tests were carried out at a constant velocity of 1 mm/min according to the ASTM D3163-01 standard:(1)$$\tau_{\max}=\frac{F_{\max}}{A}\;(\mathrm{MPa})$$
where: τ_max_ is the maximum shear stress [MPa], F_max_ is the force causing damage to the joint (N), and the A is the cross-sectional area of the joint (mm^2^).

Adhesive joints were made for the test according to the scheme shown in Figure 2. Two ceramic plates were placed in a Teflon mold. A layer of elastomer was placed between them. The thickness of the elastomer layer corresponded to the thickness of the spacer plate. The ceramic specimens with dimensions of 5 *×* 7 *×* 37 mm were first placed in a molder, and then the reactive mixture was poured.

The character of joint failure (adhesive or cohesive) and the microstructure of composites were characterized using Scanning Electron Microscopy (SEM) TM3000 (HITACHI High-Technologies Corporation, Tokyo, Japan) operating at an applied voltage of 5 kV. Before observations with the SEM, the surfaces of the specimens were sputtered with a gold-palladium layer for 90 s at a current of 10 mA and voltage of 2 kV. SkyScan 1174 X-ray tomography (SkyScan, Aartselaar, Belgium) was used for testing of the ceramic preform and composites. Before scanning, samples in the shape of a cuboid with dimensions 10 *×* 10 *×* 15 mm did not require any special preparation. Scanning was performed using an X-ray tube with the following parameters: 100 kV voltage, 100 kA, no filter material, 0.5° rotation step in an angle interval of 180°. The obtained cross-sections of the ceramic preforms and composites were studied using CTAn software v.1.18 (Bruker, Kontich, Belgium) and as a result, porosity of the ceramic preform as well as the residual porosity of composites were determined. The application of CTAn software enabled the investigation of the weight fraction of each phase, including voids in ceramic preforms as ceramics porosity and in composites as residual porosity.

The compressive test was carried out using the MTS Q/Test 10 testing machine test machine (MTS Testing Systems, Toronto, ON, Canada) according to the ISO 20504:2019 standard with 1 mm/min velocity. Based on the obtained stress–strain curves, compressive strength and stress at the plateau were calculated.

## 3. Results and Discussion

### 3.1. Surface Roughness Test of Dense Alumina Ceramics

The topography of the ceramic surface was analyzed by the results of the roughness test. Results of Ra and Rz roughness parameters, as well as a surface profile observation, are displayed in Figure 3, Figure 4 and Figure 5. The surface roughness test was conducted for dense Al_2_O_3_ ceramics fabricated by two methods; for Al_2_O_3_-SS samples, pressureless sintering was the final process, whereas for Al_2_O_3_-HIP200 samples hot isostatic pressure sintering was applied additionally.

The roughness parameters determined with the use of a laser profilometer allowed the assessment of the degree of surface development of both types of ceramic materials. It was demonstrated that the use of hot isostatic pressure during the sintering of alumina ceramics modified their surface condition. It should be noted that the application of 200 MPa pressure during ceramic sintering caused a decrease in roughness parameters in comparison to the pressureless sintered material. Looking into the ceramic profiles, in the Al_2_O_3_-HIP200 sample, numerous structural defects with a size of about 50 μm and a depth of 9 μm were eliminated. Other researchers have previously observed a similar relationship in alumina ceramic sintering [24]. In comparison, Al-Jawoosh et al. indicated that the R_a_ roughness parameter for the densely sintered alumina ceramic specimens was 0.6 µm [25]. As a consequence of the use of high pressure, the pores and cracks closing occurred and the material’s compaction increased. Furthermore, densification was due to particle rearrangement, plastic deformation, grain boundary diffusion, and structural defect elimination. During the sintering of a powder compact, both densification and grain growth occur simultaneously [26].

### 3.2. The Contact Angle Measurements of Dense Alumina Ceramics

The contact angle measurements were performed to examine the effect of the HIP sintering application as well as the silane coupling agent on ceramic wettability. Furthermore, the influence of the coupling agent content in the toluene solution on the contact angle was investigated. The results of the tests are presented in Figure 6.

Wettability is the ability to spread material in a liquid form on a solid surface. Depending on the contact angle value, surfaces are of a hydrophilic character (θ < 90°) or hydrophobic character (θ > 90°). In the case of IPC composites, the value of the contact angle affects the degree of filling of the porous preform by the liquid material, and thus affects the mechanical strength of the composite [27]. Reducing the contact angle value, i.e., improving wettability, can ensure better filling of pores in the ceramic preform, and elimination of gas bubbles from the interface. In addition, the elimination of structural discontinuities increases the contact area between the materials, which in turn leads to an improvement of the durability of the interface connection [28].

The obtained results show that the average water contact angle of the nontreated ceramic surface was about 80° (lack of differences between the fabrication methods of ceramic) but in the presence of a solution coat of 5 wt% U-15 coupling agent in toluene, it was decreased to 60°. This means that the wettability of ceramic was improved due to the U-15 agent application. Although an angle value below 90° indicates the hydrophilic character of the ceramic surface uncoated by the solution of the U-15 agent, the contact angle was still close to 90°. It decreased significantly only after applying the coupling agent coat.

The solution concentration of the silane coupling agent in toluene has been selected based on previous work [29]. For low concentrations of the U-15 agent (<1%), a change in water contact angle was not observed. These observations were confirmed by other researchers [16,28]. It was found that the solution concentration, solution pH, and curing conditions can affect significantly the silane bonding to an inorganic surface. Silane solution baths are usually used in low concentrations (0.01–2%) because the formation of oligomers is suppressed in dilute solutions [28].

The literature indicates that the contact angle is closely related to surface roughness [30]. It can be concluded that if the R_a_ parameter possesses a value lower than 0.5, the effect of roughness on the contact angle is insignificant. Higher R_a_ values indicate an increase in surface roughness, hence, an increase in the wetting surface (contact surface). Nevertheless, increasing roughness can affect wettability in two ways. Together with the improvement in surface roughness, the number of defects and pores on the surface into which liquid material can penetrate increases. As a result, the strength of such a joint can enhance. However, too many pores, especially narrow ones, become an obstacle, impede wettability, and consequently, prevent the formation of a durable adhesive interface connection [30]. As shown in Figure 2, the achieved Ra roughness parameter is low independently of the fabrication method of dense alumina ceramic. This confirms the lack of a significant difference in water contact angle results for Al_2_O_3_-SS and Al_2_O_3_-HIP200 ceramic.

### 3.3. Shear Strength of an Interface Joint

The shear strength results of the ceramic–elastomer interface joint are displayed in Figure 7. It can be seen that the application of the silane solution coat enhanced the shear strength of the ceramic–elastomer joints. The average shear strength value for Al_2_O_3_-SS ceramic–elastomer joint was 2.9 ± 0.7 MPa while using a silene solution coat increased the shear strength to 8.8 ± 0.2 MPa. In comparison, Chaijareenont et al. indicated that silane coupling agents affect polymethyl methacrylate (PMMA) bonding to alumina. The bond strength of PMMA on the alumina treated by a bath in a solution of N-2 (aminoethyl) 3-aminopropyltriethoxysilane) in ethanol was reached at 10.8 MPa, which is similar to obtained results [31]. The highest shear strength was achieved for the ceramic–elastomer joint in which dense ceramic was a fabrication by pressureless sintering. This is related to the slightly higher roughness parameters of the surface specimen. In other words, a higher surface roughness caused an increase in the contact area of the joint materials, while the use of a coupling agent ensured the possibility to infiltrate the micropores on the ceramic surface by the elastomer reactive mixture. It is noteworthy that, in the case of dense ceramic fabricated by pressureless sintering, the pore size was not reduced as for Al_2_O_3_-200HIP samples. Hence, the penetration of the reactive mixture into the micropores of the ceramic surface and the creation of additional mechanical bonds was easier. In addition, due to its bi-functionality, the U-15 agent interacted with two materials, improving their mutual adhesion by creating molecular bridges, which connected the inorganic surface with the polymer through the available types of polar interaction. The bonding was covalent (siloxane bond) [32].

### 3.4. SEM Observations of Ceramic–Elastomer Joint

The ceramic–elastomer bond quality was analyzed in terms of failure character joint after a shear test using a scanning electron microscope. Sample images are presented in Figure 8a,b. At first glance, it can be seen that the failure was adhesive for interfacial joints in which no coupling agent was applied, as evidenced by the smooth surface of the joint after the shear test. In contrast, in samples in which the coupling agent was applied (Figure 8b), the failure was reached partially by elastomer decohesion, which indicated good adhesion of the elastomer to the ceramic surface.

### 3.5. Mechanical Properties of Porous Alumina Ceramic

The main property of porous ceramic preforms is high mechanical strength. Therefore, the effect of the fabrication method as well as the degree of porosity of ceramic preform into their compressive strength were analyzed (Table 1). The results show that the average compressive strength of the sample fabricated by pressureless sintering was about 10 MPa, but with the addition of the HIP sintering stage, it was increased by 100%. This is related to the densification of alumina grains and pores size reduction.

Ceramic preforms intended for pressure infiltration should exhibit mechanical strength. Otherwise, they may be damaged during the fabrication of composites. Looking into the compressive test result, it can be seen that porous preform fabricated using HIP sintering is characterized by higher mechanical strength.

Interestingly, the degree of porosity of ceramic preform achieved in the work is considerably lower in comparison to ceramic preform porosity that can be obtained using other methods. In comparison, Peng et al. indicated that the porosity of ceramic foams fabricated by the polymeric sponge method was higher than 70% [33]. Similarly, the application of gel casting of ceramic foams method allowed porosity to be reached between 50% and 90% [34]. However, it is noteworthy that the growth in porosity causes a decrease in mechanical strength.

### 3.6. SEM Observations of Composites

After characterization of porous ceramic preforms’ mechanical properties, a part of them was used for ceramic–elastomer composites fabrication. The mechanical test results of porous samples confirmed that the HIP sintering application allowed them to achieve higher compressive strength. From this point of view, the porous ceramic preforms were utilized for ceramic–elastomer fabrication by the infiltration method. The Al_2_O_3_-HIP200 porous samples with 20% and 40% porosity were infiltrated by the liquid elastomer. Part of the samples was coated with 5 wt% silane solution in toluene. To assess the microstructure of the fabricated ceramic–elastomer composites and the effect of the coupling agent on their microstructure, SEM observations were conducted. The obtained results are shown in Figure 9b and Figure 10b. In contrast, observations of composite fabricated by using porous ceramic preform without a U-15 agent solution coat are displayed in Figure 9a and Figure 10a.

At first glance, it can be seen that the macro and micro-pores of the ceramic preforms have been filled by the reactive mixture of elastomer. The elastomer also infiltrated the channels formed between the ceramic grains, as well as in their cracks. It should be also noticed that the structure of interpenetrating phases was successfully obtained. Observations confirmed that the infiltration method allowed the elastomer to fill the ceramic pores. Looking into the SEM images, it can be seen that the adhesion between Al_2_O_3_ ceramics with the elastomer for each porosity was improved for composites fabricated by using porous ceramic preform with a U-15 agent solution coat. Furthermore, the pores have been filled better and the interface boundary between the ceramics and the elastomer was continuous. In addition, delamination on the ceramic–elastomer boundary was not observed, contrary to composites obtained using porous ceramic preform without a U-15 agent solution coat. In the case of uncoated samples, the effect was revealed of weaker adhesion between the ceramic and polymer phases, as evident from the isolated debonds highlighted in Figure 9a and Figure 10a. It can be concluded that the silane coupling agent facilitates infiltration and improves adhesion between the phases; other researchers have previously observed a similar relationship by coupling agent application [35,36,37].

It is noteworthy that after the ceramic pores are filled with elastomer in liquid form, polymerization of the monomer is started, and the unavoidable volume shrinkage can appear. In work [10], it was confirmed that the application of pressure during infiltration as well as a reduced polymerization speed can limit the occurring polymerization shrinkage, which leads to a decrease in the appearance of defects in the microstructure of IPCs. In this study, despite pressure infiltration application to fill ceramic pores with liquid elastomer, the polymerization shrinkage generated pores (defects) in the microstructure. This led to an interfacial boundary loss between polymer and ceramic. An optimized, defect-free microstructure was obtained for composites fabricated by infiltration of ceramic preforms coated by silane agent.

### 3.7. Residual Porosity Measurement of Composites

Residual porosity, i.e., porosity, which is a result of insufficient pore filling by the elastomer, was determined by X-ray tomography. The results of residual porosity for composites fabricated by infiltration of porous ceramic pre-form coated with a U-15 agent solution as well as uncoated ceramic preforms are shown in Figure 11.

The coupling agent had a significant impact on the infiltration process and, consequently, the degree of pore filling. The smallest residual porosity after infiltration, approximately 2 vol.%, was evaluated for composites fabricated by infiltration of ceramic pre-form with 40 vol.% porosity coated with a U-15 agent solution. Infiltration of preforms with higher porosity was easier, pores were better filled, and residual porosity was smaller than in the case of the preform with smaller porosity.

### 3.8. Mechanical Properties of Composites

To analyze how the coupling agent application affects the compressive strength and stress at the plateau of composites, the compression test was performed. The calculated compressive strength and stress at the plateau area are presented in Figure 12 and Figure 13. For the composites fabricated by infiltration of the uncoated porous preform, the compressive strength was under 30 MPa. Using silane solution on porous ceramic preforms increased the compressive strength of the composite to about 35 MPa. A similar tendency of stress in the plateau area growth was found. The obtained results can be summarized: the application of the U-15 coupling agent caused a significant increase in the mechanical properties of ceramic–elastomer interpenetrating phase composites. In addition, the higher score of the area under the stress-strain curve, i.e., the plateau area, as well as stress at the plateau, was achieved. Most likely, the ability to absorb the energy of composites fabricated with a U-15 promoter application was also improved. Moreover, the samples were not destroyed during the compressive test. After removing the load, the composite specimens almost returned to their original shape, because of the highly elastic deformations of the elastomer. Similar results were obtained in [18], where the increase in elastic modulus and compressive strength of silane-coated preform can be attributed to improved wettability, which in turn enhances adhesion between the metal and polymer phases.

Figure 14 shows the typical stress–strain response at static compressions of IPCs. For all composites coated and untreated with silane solution, a linear elastic deformation followed by a protracted nonlinear behavior was observed. After reaching the elastic limit, the inelastic stage includes a distinct softening response due to the onset of ceramic foam failure. It is important to note that the plateau stage is characterized by a long duration of slightly increased stress and quickly increased strain. The last stage is concerned with the densification of the composite structure due to load impact. As noted earlier, the increase in compressive strength and stress of plateau of composites fabricated by using silane-coated preform can be attributed to improved wettability, which in turn enhances adhesion between the ceramic and polymer phases. The improvement in the silane-coated composites’ mechanical characteristics relative to the uncoated one is caused by the stronger bond between the ceramic and polymer matrix. Application of silane coupling agent delays failure of interfacial bonds during the deformation process [17,18].

## 4. Conclusions

This paper describes the influence of surface modification and fabrication methods of alumina ceramic on the adhesion between ceramic and elastomer in interpenetrating phase composites. The application of the HIP process for the fabrication of ceramic foams is rare, due to the difficulty of gaining IPCs afterward. Owing to the use of HIP, a low degree of porosity of the ceramic preforms was achieved, which provides higher mechanical properties of the composites. However, for such porosity, it is difficult to obtain a composite structure with interpenetrating of phases. Hence, silane coat applications were analyzed. The tests were carried out for dense and porous alumina ceramic as well as alumina–elastomer composites. The porous ceramic preforms were obtained by sintering methods whereas composites used infiltration of reactive elastomer mixture into ceramic pores. The fabricated composites were characterized by the microstructure with two three-dimensional and topologically interpenetrating phases. To improve the adhesion between alumina ceramics and urea-urethane elastomer in IPCs, hot isostatic pressing of alumina ceramic was utilized. Moreover, the porous ceramic preform was coated by coupling agent solution. The impact of HIP sintering and silane solution coat was investigated on roughness, wettability, microstructure, as well as mechanical properties of materials based on alumina ceramic.

The results are summarized as follows:
The application of hot isostatic pressure in the fabrication process of solid alumina ceramic affects slightly the roughness of the ceramic surface; however, in the case of porous ceramic, preforms’ mechanical strength increases by over 100%, from 10.2 MPa to 21.3 MPa for samples with 20% porosity fabricated by HIP application.Silane solution coat application achieved a significant improvement in the ceramic surface’s wettability. The water contact angle was reduced from 80% to 60%.A decrease in the contact angle facilitated the infiltration process of the porous ceramic preform by the elastomer. As a result, the degree of filling of the pores by the elastomer reactive mixture was enhanced for the ceramic–elastomer composite. Moreover, the residual porosity of composites decreased to 2%.The mechanical properties of composites, such as compressive strength and stress at the plateau increased for composites fabricated using porous ceramic coated by the solution of coupling agent. As a result of using the silane coat, compressive strength, as well as stress at a plateau, increased by more than 20%, from 25 MPa to 33 MPa and from 15 MPa to 24 MPa, respectively, for the composites fabricated by infiltration ceramic preforms with 40% of porosity.


## Figures and Tables

**Figure 1 materials-15-07824-f001:**
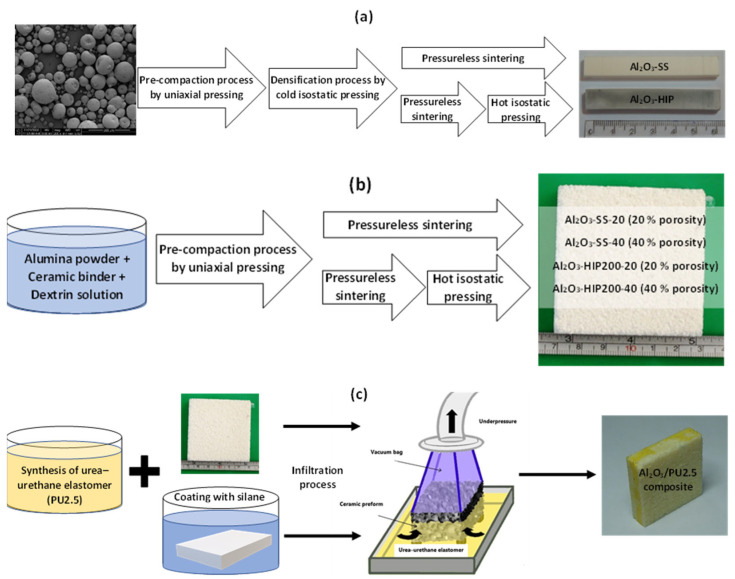
Scheme of materials fabrication processes: (**a**) dense ceramics fabrication, (**b**) porous ceramics preform production, (**c**) composite fabrication, respectively.

**Figure 2 materials-15-07824-f002:**
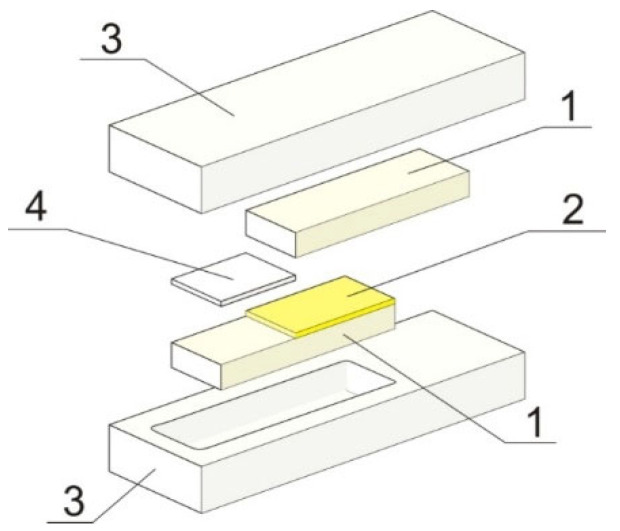
Graph showing the construction of the adhesive joint, 1—dense ceramic substrate, 2—elastomer, 3—mold, 4—spacer.

**Figure 3 materials-15-07824-f003:**
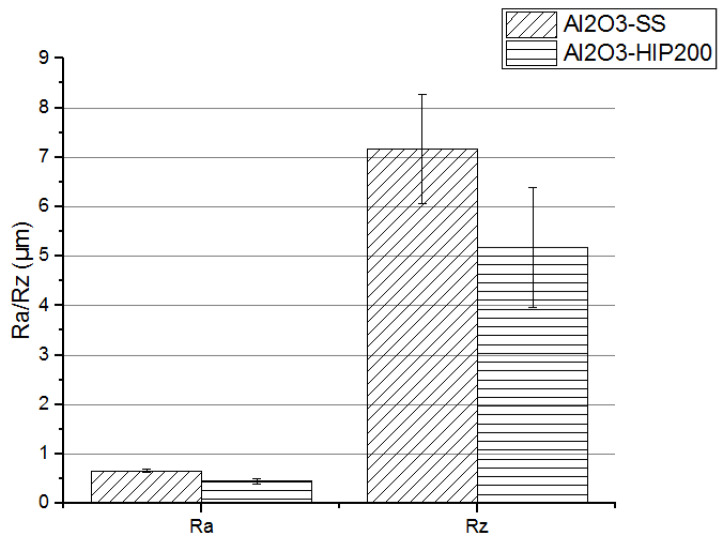
The R_a_ and R_z_ roughness parameters of dense alumina ceramics depend on fabrication methods: Al_2_O_3_-SS samples fabricated by pressureless sintering and Al_2_O_3_-HIP200 specimens obtained using hot isostatic pressure sintering.

**Figure 4 materials-15-07824-f004:**
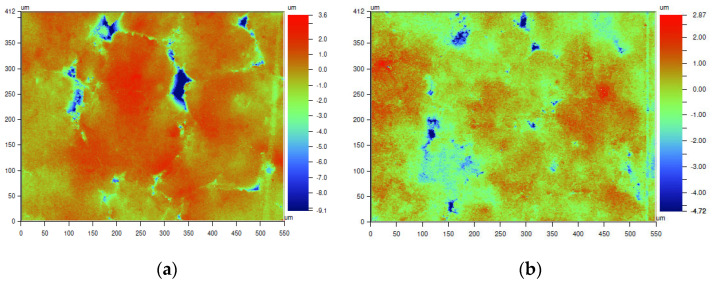
The surface topography map of dense alumina ceramics depends on fabrication methods: (**a**) Al_2_O_3_-SS samples fabricated by pressureless sintering and (**b**) Al_2_O_3_-HIP200 specimens obtained using hot isostatic pressure sintering.

**Figure 5 materials-15-07824-f005:**
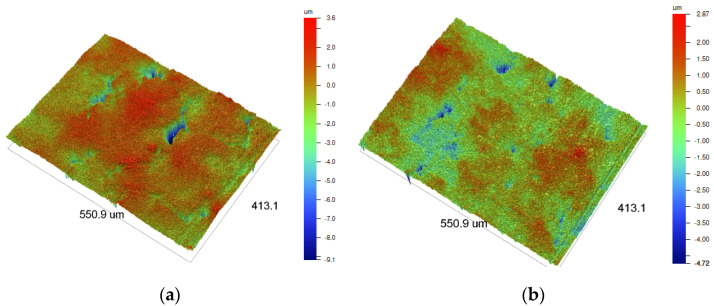
3D Profilometry images illustrating dense alumina ceramics depending on fabrication methods: (**a**) Al_2_O_3_-SS samples fabricated by pressureless sintering and (**b**) Al_2_O_3_-HIP200 specimens obtained using hot isostatic pressure sintering.

**Figure 6 materials-15-07824-f006:**
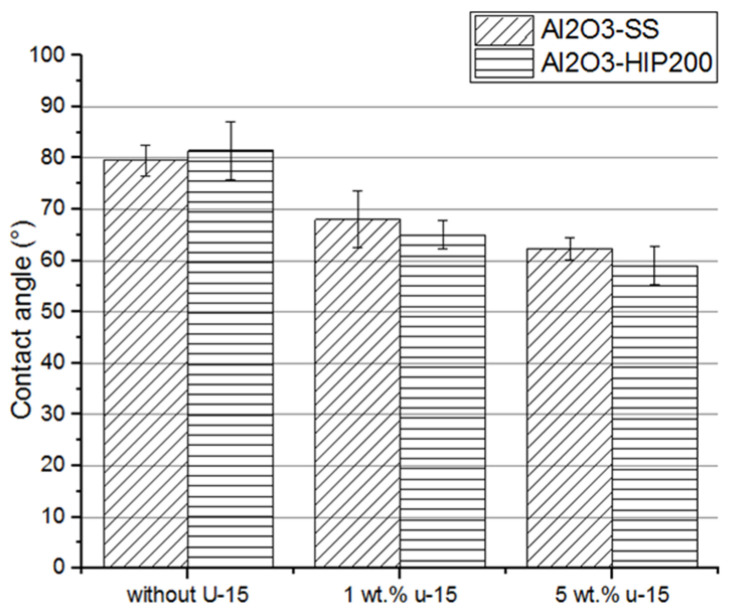
The contact angles on the alumina ceramic surface were treated using a silane solution of the coupling agent with different concentrations; Al_2_O_3_-SS samples were fabricated by pressureless sintering, and Al_2_O_3_-HIP200 specimens were obtained using HIP sintering.

**Figure 7 materials-15-07824-f007:**
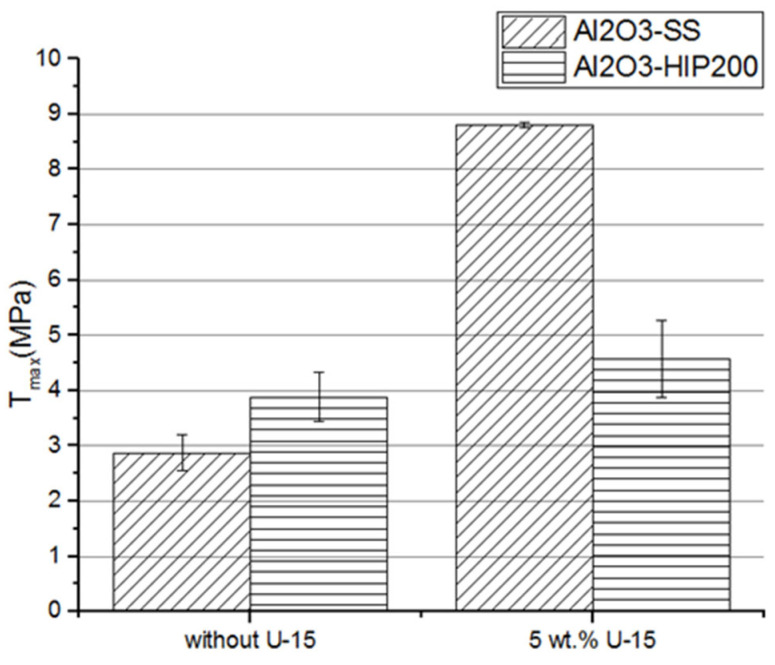
The shear strength on the alumina ceramic surface was treated using a silane solution of the coupling agent with different concentrations; Al_2_O_3_-SS samples were fabricated by pressureless sintering, and Al_2_O_3_-HIP200 specimens were obtained using HIP sintering.

**Figure 8 materials-15-07824-f008:**
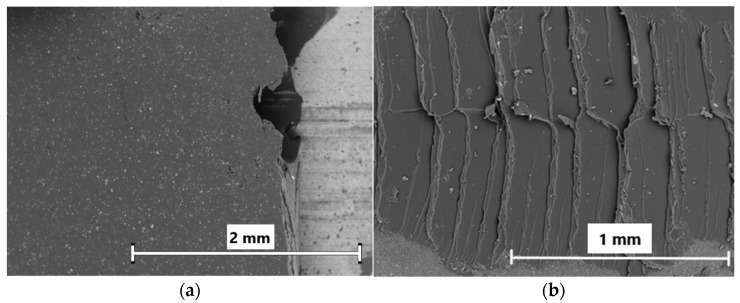
SEM images illustrating failure after shear strength test of ceramic–elastomer joints: (**a**) alumina surface uncoated of silane solution, (**b**) alumina surface coated of silane solution.

**Figure 9 materials-15-07824-f009:**
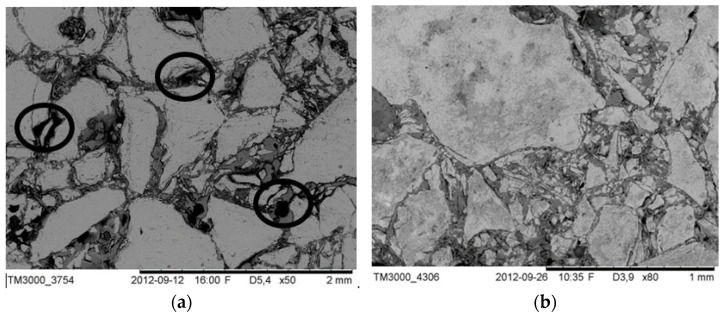
The microstructure of Al_2_O_3_/PU2.5 composite fabricated by infiltration of the ceramic preform with 20% porosity: (**a**) porous alumina surface uncoated of silane solution, (**b**) porous alumina surface coated of silane solution. (The lighter phase is alumina ceramics and the darker phase is polyurethane elastomer, the black circle show structural defects).

**Figure 10 materials-15-07824-f010:**
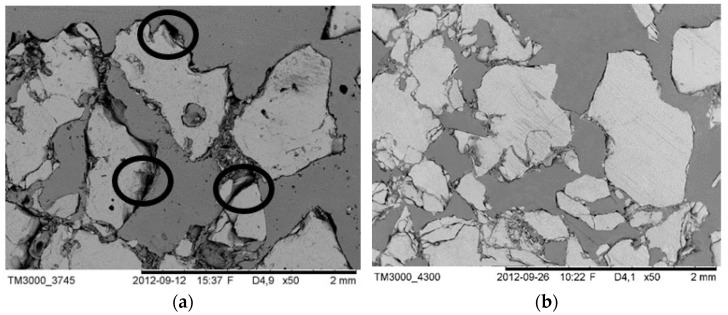
The microstructure of Al_2_O_3_/PU2.5 composite fabricated by infiltration of the ceramic preform with 40% porosity: (**a**) porous alumina surface uncoated of silane solution, (**b**) porous alumina surface coated of silane solution. (The lighter phase is alumina ceramics and the darker phase is polyurethane elastomer, the black circle show structural defects).

**Figure 11 materials-15-07824-f011:**
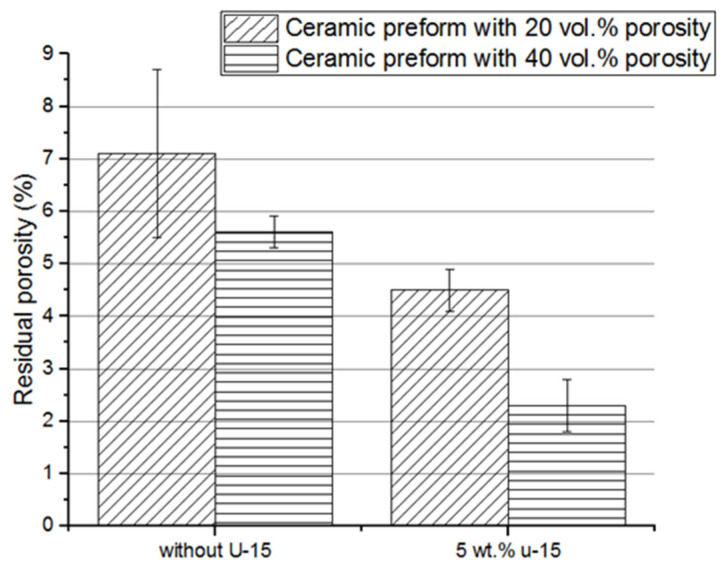
Comparison between residual porosity of composites fabricated by using porous alumina uncoated and coated of silane solution; Al_2_O_3_/PU2.5 composites fabricated by infiltration of the ceramic preform with 20% and 40% porosity.

**Figure 12 materials-15-07824-f012:**
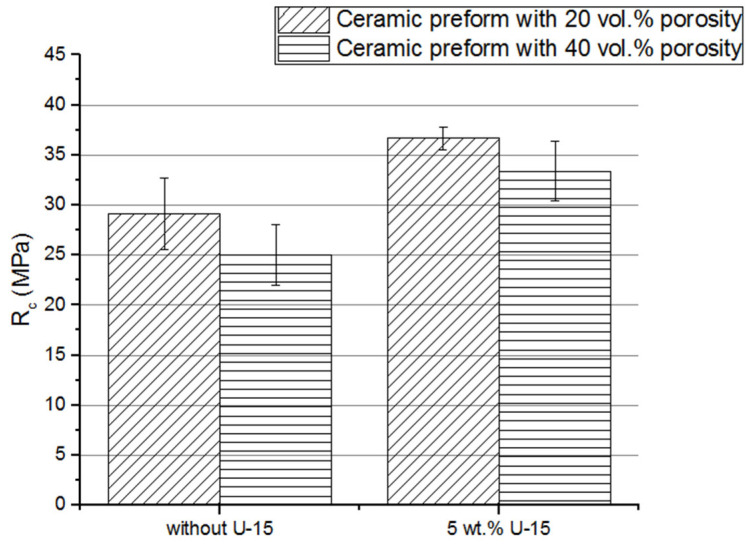
Comparison between compressive strength of composites fabricated by using porous alumina uncoated and coated of silane solution; Al_2_O_3_/PU2.5 composites fabricated by infiltration of the ceramic preform with 20% and 40% porosity.

**Figure 13 materials-15-07824-f013:**
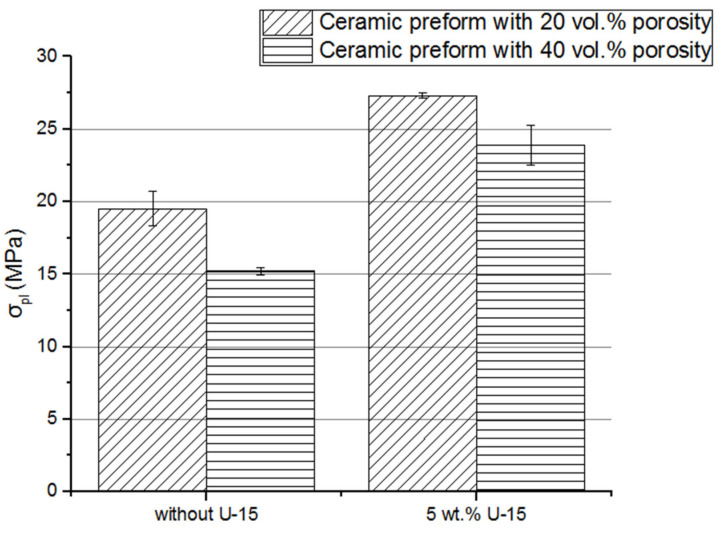
Comparison between stress at a plateau of composites fabricated by using porous alumina, uncoated and coated, of silane solution; Al_2_O_3_/PU2.5 composites fabricated by infiltration of the ceramic preform with 20% and 40% porosity.

**Figure 14 materials-15-07824-f014:**
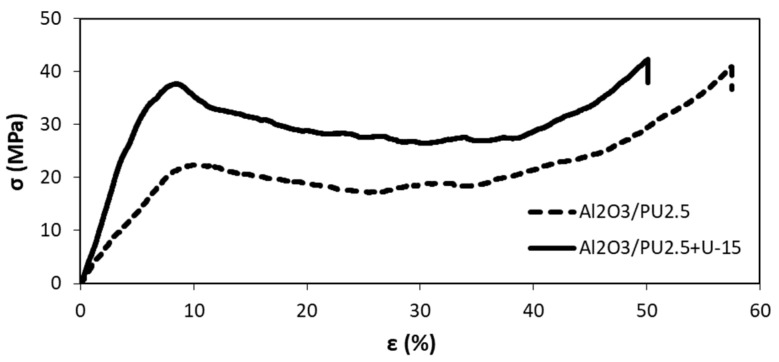
Stress–strain response in uniaxial compression for composites with uncoated (Al_2_O_3_/PU2.5) and silane coated performs (Al_2_O_3_/PU2.5+U-15); ceramic foam with 20% porosity.

**Table 1 materials-15-07824-t001:** Compressive strength of porous ceramic preform fabricated by pressureless and HIP sintering.

Material	Type of Applied Final Stage of a Fabrication Method	Porosity [%]	Compressive Strength [MPa]
Al_2_O_3_-SS-20	Pressureless sintering	22 ± 0.5	10.2 ± 2.1
Al_2_O_3_-SS-40		41 ± 1.0	6.5 ± 1.2
Al_2_O_3_-HIP200-20	Hot isostatic pressing	23 ± 0.7	21.3 ± 3.0
Al_2_O_3_-HIP200-40		40.5 ± 2.0	17.6 ± 1.4

## Data Availability

All data are included in the manuscript.

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
