# Peer review of "Influence of Fabrication Method and Surface Modification of Alumina Ceramic on the Microstructure and Mechanical Properties of Ceramic–Elastomer Interpenetrating Phase Composites (IPCs)"

_materials, 2022, doi:10.3390/ma15217824_

Round 1

Reviewer 1 Report

In this manuscript, the influence of surface modification and fabrication methods on the microstructure and mechanical properties of ceramic-elastomer composites have been investigated. The content of this article is substantial and well organized. Hence, this paper could be accepted after addressing the following issues.

1.     Please revise the sections of abstract and conclusion, which are more concise and readable.

2.     Nacre-mimetic composites show similar structure of ICPs and improved mechanical properties(Advanced Materials 2020, 32, 1907176), which are fabricated by using a preformed scaffold infiltrating with a second phase. It is suggested to discussed in the section of Introduction.

3.     The mechanism of the improved mechanical properties should be further discussed.

4.     Please check the part of Figure 2 on the top right.

Author Response

Point 1: Please revise the sections of abstract and conclusion, which are more concise and readable.

Response 1: Thank You for this comment. The abstract and conclusion were revised.

Point 2: Nacre-mimetic composites show similar structure of ICPs and improved mechanical properties(Advanced Materials 2020, 32, 1907176), which are fabricated by using a preformed scaffold infiltrating with a second phase. It is suggested to discussed in the section of Introduction.

Response 2: Thank You for this comment. Introduction updated.

Point 3: The mechanism of the improved mechanical properties should be further discussed.

Response 3: Thank You for this comment. The explanation was added.

Point 4: Please check the part of Figure 2 on the top right.

Response 4: Thank You for this comment. Figure corrected.

Reviewer 2 Report

1. in general, i believe this paper is not well orgnized.  for instance, the authors should at least provide a figure to demostrate the experimental procedure of their works rather than tedious descrption.

2. the quality of figures in this study must be improved, such as figure.2.

3. the authors need to discuss their data fully instead of just plot it without deep expalnation.

4. how does the porosity be measured?

5. In figure 8 and 9, the authors claim that stronger adhension is obtained between ceramic and polymer phases after coating. in fact, no obvious evidence can be indicated in SEM pictures. i believe the solid evidence should be provided.

6.since the authors mentioned ceramic-polymer materials, the following articles might be relevant:

Cannillo V, Bondioli F, Lusvarghi L, et al. Modeling of ceramic particles filled polymer–matrix nanocomposites[J]. Composites science and technology, 2006, 66(7-8): 1030-1037.

Shao J, Zhou L, Chen Y, et al. Model-based dielectric constant estimation of polymeric nanocomposite[J]. Polymers, 2022, 14(6): 1121.

Kromer R, Danlos Y, Aubignat E, et al. Coating deposition and adhesion enhancements by laser surface texturing—metallic particles on different classes of substrates in cold spraying process[J]. Materials and manufacturing Processes, 2017, 32(14): 1642-1652.

7.the novelties in this paper should be clearly indicated, as so many ceramic-polymer materials have been reported.  

Author Response

Point 1: in general, i believe this paper is not well orgnized. for instance, the authors should at least provide a figure to demostrate the experimental procedure of their works rather than tedious descrption.

Response 1: Thank You for this comment. A scheme was added.

Point 2: the quality of figures in this study must be improved, such as figure.2.

Response 2: Thank You for this comment. Figure corrected.

Point 3: the authors need to discuss their data fully instead of just plot it without deep expalnation.

Response 3: Thank You for this comment. The explanations were added.

Point 4: how does the porosity be measured?

Response 4: Thank You for this comment. The explanation was added.

The SkyScan 1174 X-ray tomography (SkyScan, Aartselaar, Belgium) was used to testing of ceramic preform and composites. Before scanning, samples in the shape of a cuboid with dimensions 10 x 10 x 15 mm did not require any special preparation. Scanning was performed using an X-ray tube with the following parameters: 100 kV voltage, 100 kA, no filter material, 0.5 ° rotation step in an angle interval of 180 °. The obtained cross-sections of the ceramic preforms and composites were studied using CTAn software (Billerica, MA, USA) and as a result, porosity of ceramic preform as well as the residual porosity of composites were determined.

Point 5: In figure 8 and 9, the authors claim that stronger adhension is obtained between ceramic and polymer phases after coating. in fact, no obvious evidence can be indicated in SEM pictures. i believe the solid evidence should be provided.

Response 5:  Thank You for this comment. The explanation was added.

Point 6: since the authors mentioned ceramic-polymer materials, the following articles might be relevant:

Cannillo V, Bondioli F, Lusvarghi L, et al. Modeling of ceramic particles filled polymer–matrix nanocomposites[J]. Composites science and technology, 2006, 66(7-8): 1030-1037.

Shao J, Zhou L, Chen Y, et al. Model-based dielectric constant estimation of polymeric nanocomposite[J]. Polymers, 2022, 14(6): 1121.

Kromer R, Danlos Y, Aubignat E, et al. Coating deposition and adhesion enhancements by laser surface texturing—metallic particles on different classes of substrates in cold spraying process[J]. Materials and manufacturing Processes, 2017, 32(14): 1642-1652.

Response 6: Thanks for the suggestion, the articles have been analyzed.

Point 7: the novelties in this paper should be clearly indicated, as so many ceramic-polymer materials have been reported.

Response 7: Thank You for this comment. The conclusion was corrected.

Reviewer 3 Report

The article presents the results of a study of the influence of technological parameters on the properties of ceramic-elastomer interpenetrating phase composites. The work was done at a high professional level and can be published after making changes, taking into account some comments:

- The description of figures 8 and 9 requires a more thorough explanation and characterization of the presented microstructure. It's not clear what the "darker" gray area represents between the ceramic particles? Compound for preparing polished samples? For a more thorough analysis of mechanical characteristics and microstructure, there is not enough data on the average grain size, or grain size distribution after sintering.

- Figures 11 and 12 are presented with only one dependence, and do not contain illustrations "a" and "b". Figure captions need to be corrected.

- A lot of spaces are missing between the value and units, for example lines 32, 84, 193, 194 ... 508, 519, 520, etc.

Author Response

Point 1: The description of figures 8 and 9 requires a more thorough explanation and characterization of the presented microstructure. It's not clear what the "darker" gray area represents between the ceramic particles? Compound for preparing polished samples? For a more thorough analysis of mechanical characteristics and microstructure, there is not enough data on the average grain size, or grain size distribution after sintering.

Response 1: Thank You for this comment. The description was updated. After the fabrication of ceramic preforms, we focused on determining the porosity and investigating the correlation with the results of mechanical tests.

Point 2: Figures 11 and 12 are presented with only one dependence, and do not contain illustrations "a" and "b". Figure captions need to be corrected.

Response 2: You for this comment. Figure captions were improved.

Point 3: A lot of spaces are missing between the value and units, for example lines 32, 84, 193, 194 ... 508, 519, 520, etc.

Response 3: Thank You for this comment. The text was improved.

Round 2

Reviewer 1 Report

This manuscript can be accepted in current form

Reviewer 2 Report

Accept in present form